# Genome-Wide Identification and Posttranscriptional Regulation Analyses Elucidate Roles of Key Argonautes and Their miRNA Triggers in Regulating Complex Yield Traits in Rapeseed

**DOI:** 10.3390/ijms24032543

**Published:** 2023-01-28

**Authors:** Liyuan Zhang, Bo Yang, Chao Zhang, Huan Chen, Jinxiong Xu, Cunmin Qu, Kun Lu, Jiana Li

**Affiliations:** 1Chongqing Rapeseed Engineering Research Center, College of Agronomy and Biotechnology, Southwest University, Chongqing 400716, China; 2Academy of Agricultural Sciences, Southwest University, Chongqing 400716, China

**Keywords:** miRNA–Argonautes, phylogenetic and expression analysis, posttranscriptional regulation, *Brassica napus*, yield-related traits

## Abstract

Argonautes (AGOs) interact with microRNAs (miRNAs) to form the RNA-induced silencing complex (RISC), which can posttranscriptionally regulate the expression of targeted genes. To date, however, the AGOs and their miRNA triggers remain elusive in rapeseed (*Brassica napus*). Here, we systematically performed a phylogenetic analysis and examined the collinear relationships of the AGOs among four Brassicaceae species. Their physicochemical properties, gene structures, and expression patterns among 81 tissues from multiple materials and developmental stages were further analyzed. Additionally, their posttranscriptional regulation was analyzed using psRNATarget prediction, miRNA-/mRNA-Seq analyses, and a qRT-PCR verification. We finally identified 10 AtAGOs, 13 BolAGOs, 11 BraAGOs, and 24 BnaAGOs. An expression analysis of the *BnaAGO*s in the *B. napus* cultivar ZS11, as well as genotypes with extreme phenotypes in various yield-related traits, revealed the conservation and diversity of these genes. Furthermore, we speculated the posttranscriptional regulation of the *B. napus* miR168a–AGO1s and miR403–AGO2s modules. Combining miRNA-Seq and mRNA-Seq analyses, we found that the *B. napus* miR168a–AGO1s module may play an essential role in negatively regulating yield traits, whereas the miR403–AGO2s module positively impacts yield. This is the first attempt to comprehensively analyze the AGOs and their miRNA triggers in *B. napus* and provides a theoretical basis for breeding high-yielding varieties through the manipulation of the miRNA–AGOs modules.

## 1. Introduction

MicroRNAs (miRNAs) are multifunctional and endogenous small nucleotide molecules, which are just 20–24 nucleotides in length but play crucial roles in the posttranscriptional regulation of the expression of target genes [1,2]. They are pivotal components of the complex network-regulating plant development, stress tolerance, and hormone signaling [3,4]. The properties of one miRNA can regulate multiple paralogous genes simultaneously, making them a research hotspot for complex quantitative traits. Many previous studies have elucidated that miRNAs induce pleiotropic effects in the regulation of complex yield-related traits in crops; for instance, functional analyses of transgenes in rice (*Oryza sativa* L.) revealed that *miR396* can increase grain yields by enhancing grain size and panicle branching, and simultaneously altering the plant architecture [5,6]. The elevated expression of *miR397* also promotes panicle branching and grain enlargement by targeting *OsLAC*, leading to an increase in grain yield of up to 25% [7]. Similarly, the overexpression of *miR408* enhanced the grain yield in rice by increasing the grain number and panicle branches [8]. Many other reports also showed that miRNAs can simultaneously alter multiple traits by targeting different paralogous genes [9,10,11]. In order to execute their functions, all miRNAs require a physical interaction with the *Argonaute* (*AGO*) genes to form the RNA-induced silencing complex (RISC) [12,13], which leads to the cleavage of the target mRNA or translational repression.

Argonautes are encoded by multiple gene families and different *AGO* genes are present in different species [14]. Abundant *AGO* genes have been identified in several plant species, including 10 AGOs in *Arabidopsis thaliana* [15], 18 in rice [16], 18 in maize (*Zea mays* L.) [17], 21 in sugarcane (*Saccharum spontaneum* L.) [18], and 15 in tomato (*Solanum lycopersicum* L.) [19]. Among them, *AGO1* is the most well-known plant *AGO* gene, which encodes a necessary component of RISC [20]. The *Arabidopsis* AGO2 protein is essential to antiviral defense, catalyzing viral RNA cleavage [21]. AGO10 is the closest paralog of AGO1, showing functional redundancy in some aspects, but is also known to be a decoy for miR166 to prevent the formation of the AGO1–miR166 complex to finely modulate the maintenance of the shoot apical meristem by regulating the expression of the target *HD-ZIP III* genes [22,23]. AGO4, AGO6, and AGO9 recruit endogenous miRNAs for DNA methylation [24,25]. Although AGO proteins are the main component of RISC and are, therefore, involved in miRNA-mediated mRNA cleavage and posttranslational repression, some AGOs are in turn regulated by miRNAs via a feedback mechanism; for instance, miR168 targets AGO1 in many species [26,27,28], and AGO2 is regulated by miR403 [29,30,31].

Rapeseed (*Brassica napus* L.; AACC, 2n = 38) is the most important oilseed crop around the world [32]. Increasing *B. napus* yield is a major focus of crop breeders; however, the yield-related traits in *B. napus* are complex quantitative characters controlled by polygenes and environmental factors [33]. Additionally, *B. napus* is an allotetraploid species, which was formed by a cross between field mustard (*B. rapa* L.) and wild cabbage (*B. oleracea* L.) around 7500 years ago. Many studies have identified plentiful redundant genes that were not lost during the genomic evolution of allotetraploid *B. napus* [32,34,35,36]. This genomic complexity makes the functional study of individual genes particularly challenging, and most studies have focused on fundamental research of screening candidate genes that may be related to yield-related traits based on genome-wide association studies (GWAS) or quantitative trait loci (QTL) analyses [37,38,39]; the handful of functional analyses of candidate genes are limited to a single structure factor only. For these reasons, the AGOs and their miRNA triggers remain elusive in *B. napus*.

In our study, we performed a genome-wide identification of *AGOs*, and examined their evolutionary relationships in four Brassicaceae species. We systematically analyzed the protein profiles of the 24 BnaAGOs in *B. napus*. Additionally, we confirmed that four *BnaAGO1*s and three *BnaAGO2*s were regulated by *B. napus* miR168a and miR403, respectively, through psRNATarget predication, a miRNA-/mRNA-Seq analysis, and in vivo qRT-pCR verification methods. Using miRNA-Seq and mRNA-Seq data from cultivars with varying yields, we speculated that the *B. napus* modules comprising miR168a–AGO1s negatively regulate yield traits, especially the thousand-seed weight (TSW), while the miR403–AGO2s modules exhibited the opposite effect. To our knowledge, this is the first report to identify the AGOs and their miRNA triggers in *B. napus*, and consequently the first to examine their potential functions in regulating complex yield-related traits. Our study, therefore, broadens our perspective of the pleiotropic effect of miRNAs and provides a theoretical basis for breeding high-yielding crops through the manipulation of the miRNA–AGO modules.

## 2. Results

### 2.1. Genome-Wide Prediction, Phylogenetic Analysis, and Nomenclature of AGO Genes

In order to predict *AGO* genes in *B. napus* and its progenitor species (*B. rapa* and *B. oleracea*) at the genome level, we downloaded the 10 AtAGO protein sequences from *Arabidopsis thaliana* (https://www.arabidopsis.org/index.jsp, accessed on 20 June 2020). A BLAST analysis with an E-value of 0 was used to find AGO orthologs in the three Brassica species, resulting in the identification of 13 BolAGOs, 11 BraAGOs, and 24 BnaAGOs. The full-length protein sequences of the AGOs from the four species were aligned and a phylogenetic tree was constructed. The 58 AGOs were divided into four main subgroups, AGO1, ZIPPY, AGO4, and MEL1 (Figure 1), based on previous studies [28,40] and our phylogenetic analysis. The AGO1 subgroup contains the AGO1 and AGO10 homologs; AGO2, AGO3, and AGO7 comprise the ZIPPY subgroup; while the MEL1 subgroup only contains AtAGO5, BrAGO5, BoAGO5, and two BnAGO5s. The subgroup AGO4 contains the most proteins, including the homologs of AGO4, AGO6, AGO8, and AGO9 subfamilies. The 58 AGO genes identified among the four species were named based on their phylogenetic relationships and homologies with the *A. thaliana* genes (Figure 1, Appendix A).

A syntenic analysis was performed to further determine the evolutionary relationship of the AGO family among the four species. A total of 77 homologous pairs were identified among the four species (Figure 2A, Appendix A), with significantly more homologous pairs found in the A subgenome than that in the C subgenome (Figure 2B,C). All of the *AGO* copies in *B. napus* corresponded to those of the progenitors *B. oleracea* and *B. rapa*, with the exceptions that *AGO9* was missing in *B. napus* and *AGO4* was not present in the progenitors (Table 1). The 24 *BnaAGO*s were distributed across the 16 chromosomes, with the exception of chromosomes A01, C01, and C09 (Figure 2D).

### 2.2. Protein Profiles, Gene Structure and Conserved Motif Analysis of the 24 BnaAGOs

To gain more insight into the structural evolution of the 24 *BnaAGO*s, we analyzed their physicochemical properties. Their gDNA sizes were highly similar within subgroups; the AGO1 subgroup possessed the largest gDNA lengths, from 4921 (*BnaAGO10-1*) to 6740 bp (*BnaAGO1-4*), followed by the AGO4 subgroup, while the ZIPPY subgroup genes were remarkable smaller: 3169 (*BnaAGO2-3*) to 3712 bp (*BnaAGO2-4*) (Figure 3B, Table 2). Similarly, BnaAGO proteins varied from 867 (*BnaAGO6-1* and *BnaAGO6-2*) to 1086 aa (*BnaAGO1-4*). The isoelectric points (pI) of the 24 BnaAGOs were all higher than 8.82, indicating that AGOs are basic proteins (Table 2). The conserved motifs were highly similar among the 24 BnaAGOs (Figure 3A). The intron number and gene structure were conserved across each *AGO* subfamily but varied significantly among members of different subfamilies; for instance, the AGO1, AGO4, and MEL1 subgroups possess similar gene structures and intron numbers ranging from 16 to 23, whereas the ZIPPY subgroup members contain only one to five introns (Figure 3B, Table 2). The protein characteristics of members of the same subgroup were highly similar, but differed between subfamilies, suggesting that the *AGO* genes underwent frequent gene duplication and recombination during evolution.

### 2.3. Differential Expression of 24 BnaAGOs among Differently Yielding B. napus Cultivars

We downloaded RNA-Seq data from 64 tissues in the *B. napus* cultivar ZS11 [41] to analyze the expression pattern of the 24 *BnaAGO*s. As shown by the heatmap, the 24 *BnaAGO*s have their own specific expression patterns (Figure 4); for example, most genes from the *AGO6*, *AGO7*, and *AGO9* subfamilies were hardly expressed in any of the tissues. The *BnaAGO4*s exhibited the highest expression level on average, with all four members being expressed most strongly in the pistil and seed. Genes belonging to the *AGO2* and *AGO5* subfamilies had similar expression profiles, with little to no expression in the embryo. All four *BnaAGO1*s were expressed in all 64 tissues. For some subfamilies, the members displayed varying expression patterns; for example, *BnaAGO9-3* was much more strongly expressed than its two paralogs, *BnaAGO9-1* and *BnaAGO9-2*. In addition, three genes (*BnaAGO2-1*, *BnaAGO2-3*, and *BnaAGO2-4*) from the *AGO2* subfamily had similar expression profiles, but *BnaAGO2-2* was hardly expressed in any of the 64 tissues (Figure 4).

The AGOs play an important role in the posttranscriptional regulatory functions of all miRNAs, which are multifunctional factors regulating complex quantitative traits. We therefore used RNA-Seq data for the 24 *BnaAGO*s in materials with extremely high and low harvest indexes (HI; a quantitative yield-related trait) to further investigate whether AGOs were associated with the complex yield traits in *B. napus*. The RNA-Seq data from the 11 HI tissues, partially published in our previous study [42], show high expression levels only for the *BnaAGO1*s, *BnaAGO2*s, *BnaAGO4*s, and *BnaAGO5*s, while the other *BnaAGO*s were hardly expressed in almost any of the HI materials (Figure 5A). Coincidently, only the *AGO1*, *AGO2*, *AGO4*, and *AGO5* subfamilies exhibited higher expression levels in three pairs of yield-related materials, including plants with extreme phenotypes for TSW (Figure 5B), seed number per silique (SPS; Figure 5C), and initial embryotic number (IEN; Figure 5D) (all those were unpublished data, Appendix A).

Finally, among the more highly expressed *AGO1*, *AGO2*, *AGO4*, and *AGO5* subfamily members, we found that the *BnaAGO1*s and *BnaAGO4*s were most highly expressed in the high-HI materials, especially in the pericarp and stem tissues (Figure 5A; indicated with red oval). On the contrary, the *BnaAGO2*s were significantly more highly expressed in the low-HI materials (Figure 5A; indicated with black arrows). Only *BnaAGO1*s (negative) and *BnaAGO2*s (positive) were correlated with TSW between materials with high and low values in this characteristic (Figure 5B). None of the genes showed a significant difference in expression between the materials with different SPS and IEN extremes (Figure 5C,D). We therefore speculate that the *AGO1*, *AGO2*, and *AGO4* subfamilies may be associated with the complex yield traits, with the *BnaAGO1*s and *BnaAGO4*s potentially making a positive contribution to yield traits and the *BnaAGO2*s playing the opposite function, especially with the HI and TSW traits.

### 2.4. MiRNA-Mediated Posttranscriptional Regulation of the BnaAGOs

To determine whether *BnaAGO*s are regulated by *B. napus* miRNAs, the psRNATarget online prediction tool was used to search the 24 full-length *BnaAGO* sequences against the known targets of plants miRNAs [43], using a maximum expectation of 3.5. Only *B. napus* miR168a and miR403 were found to have near-perfect complementarity to seven *BnaAGO*s: all the four *BnaAGO1*s (*BnaAGO1-1*, *BnaAGO1-2*, *BnaAGO1-3*, and *BnaAGO1-4*) and three *BnaAGO2*s (*BnaAGO2-1*, *BnaAGO2-3*, and *BnaAGO2-4*, but not *BnaAGO2-2*), which are targeted by miR168a and miR403, respectively (Appendix A). The complementarity scores for *B. napus* miR168a–AGO1s and miR403–BnaAGO2s were 3 and 0, respectively, with the cleavage mode (Table 3). The *miR168* and *miR403* sequences in *B. napus*, *B. oleracea*, *B. rapa*, and *A. thaliana* were downloaded from miRBase to illuminate their evolutionary relationships, and the phylogenetic tree was constructed with MEGA 7.0 (Appendix A). We further downloaded the regulatory networks of miR168 and miR403 in the four species (https://www.mirbase.org/, accessed on 20 October 2020), and found that all seven predicted miR168 members in the four species target *AGO1* genes and have known functions in the regulation of other miRNAs, whereas no related regulatory networks were available for miR403 (Appendix A).

To clarify whether *B. napus* miR168a and miR403 can negatively regulate *BnaAGO1*s and *BnaAGO2*s in *B*. *napus*, we performed a qRT-PCR analysis of their expression patterns in eight tissues of the cultivar ZS11. We found that *miR168a* was expressed to a significantly higher level in the pericarp and the seed at 15 days after flowering (15P and 15S) than the other six tissues (the bud, flower, and the pericarp and seed at 25 or 35 days after flowering; pericarp: 25P and 35P; seed: 25S and 35S, respectively) (Figure 6A). Coincidentally, the four *BnaAGO1*s are expressed at low levels in the 15P and 15S tissues. Negative expression patterns between *B. napus miR168a* and the *BnaAGO1*s can also be observed in the other six tissues. For *miR403* and the *BnaAGO2*s, this phenomenon was even more pronounced; *miR403* was only expressed in 25S and 35S, in which their three predicted targets were hardly expressed (Figure 6B). On the contrary, the three *BnaAGO2*s (especially *BnaAGO2-3* and *BnaAGO2-4*) exhibited higher expression levels in the Bu, Fl, 15S, 15P, 25P, and 35P tissues, in which *miR403* expression was almost non-existent. The negative expression relationships of the miR168a–AGO1s and miR403–AGO2s were most obvious in the 25P, 25S, 35P and 35S tissues (Figure 6). Taken together, our findings indicate that miR168a and miR403 mediate the negative posttranscriptional regulation of the *BnaAGO1*s and *BnaAGO2*s, respectively.

### 2.5. miRNA-Seq and mRNA-Seq Analysis of miR168a–AGO1s and miR403–AGO2s in Differentially Yielding B. napus Materials

To determine whether the *B. napus* miR168a–AGO1s and miR403–AGO2s modules are associated with yield-related traits in *B. napus*, we performed a combined miRNA-Seq and mRNA-Seq analysis to explore their expression profiles in multiple genotypes with differences in yield-related traits, including TSW, SPS, and IEN (unpublished data, Appendix A). We revealed a spatiotemporal expression specificity for both *miR168a* (Figure 7A) and *miR403* (Figure 7B) between the materials with extremely high and low TSW; *miR168a* was hardly expressed in the high-TSW materials but more strongly expressed in the low-TSW materials, while the four *BnaAGO1*s showed the opposite expression pattern (Figure 7A). By contrast, *miR403* was more highly expressed in the high-TSW materials, while the expression levels of the *BnaAGO2*s showed the opposite pattern (Figure 7B). The expression levels of *miR168a* and *miR403* showed no significant difference in the cultivars with differing SPS and IEN traits, resulting in no significant differences in the expression of the *BnaAGO1*s and *BnaAGO2*s either (Figure 7C–F).

Combining the results of our RNA-Seq analysis of the 24 *BnaAGO*s in ZS11 (Figure 4); the expression analysis in the HI, TSW, SPS, and IEN materials (Figure 5); and the miRNA-Seq and mRNA-Seq analyses of the miR168a–AGO1s and miR403–AGO2s, we can draw two main conclusions. On the one hand, the RNA-Seq data show that genes from the *AGO1*, *AGO2*, *AGO4*, and *AGO5* subgroups have specific higher expressions in certain genotypes and developmental stages (Figure 4 and Figure 5), and that the *BnaAGO1*s, *BnaAGO2*s, and *BnaAGO4*s may be associated with complex yield traits in *B. napus*, particularly TSW and HI (Figure 5). On the other hand, the joint miRNA-Seq and mRNA-Seq analyses indicate that *miR168a* may negatively impact yield traits by the posttranscriptional regulation of the *BnaAGO1*s, whereas the miR403–AGO2s module plays a positive role on the complex yield traits, especially TSW (Figure 7). The negative expression patterns observed between the miRNAs and mRNAs also suggest that both miR168a–AGO1s and miR403–AGO2s are posttranscriptionally regulated, as indicated by the qRT-PCR verification results in ZS11 (Figure 6).

## 3. Discussion

### 3.1. Genome-Wide Identification, Phylogenetic Analysis of AGOs, and Their miRNA Triggers

Rapeseed (*B. napus*, AACC, 2n = 38) is an important oil crop that evolved from a heterogeneous hybridization of two diploid species, *B. oleracea* (n = 9) and *B. rapa* (n = 10). With the publication of the whole *B. napus* genome sequence in 2014 [44], the evolution and origin of rapeseed, as well as the genome-wide identification of key genes, have become research hotspots in recent years. Argonautes (AGOs) are important proteins that can interact with miRNAs to form RISCs, which play essential roles in regulating gene silencing [45]. Abundant *AGO* genes were identified in many plants [15,16,17,18,19,46]. The *BnaAGO* genes were predicted in a previous study [47]; however, the miRNA–AGO complexes have not yet been characterized. Here, we identified 24 *BnaAGO*s, as well as 13 *BolAGO*s and 11 *BraAGO*s in the progenitor species (Figure 1, Appendix A). The *AGO*s in *B. napus* corresponded to orthologs in the progenitors *B. oleracea* and *B. rapa*, except that *B. napus* had one less *AGO9* and one extra *AGO4*. By comparison, the *AGO3* and *AGO8* subfamily was smaller in both *B. rapa* and *B. oleracea* than in *Arabidopsis* (Figure 1, Table 1, and Appendix A). Taken together, the results of the *AGO* homolog prediction, phylogenetic analysis, and copy number analysis of the 24 identified *BnaAGO*s are consistent with the complex genome of *B. napus* as an allotetraploid.

As mentioned above, many redundant genes have been generated during the genomic evolution of the allotetraploid *B. napus*. The yield of this crop is a typical and complex quantitative trait [33], the improvement of which would, therefore, require altering the expression of several genes. Individual miRNAs have the ability to target multiple genes simultaneously, making them increasingly common targets for the research of complex quantitative traits. The *AGO* genes are important components of RISCs; thus, we used the psRNATarget prediction to search the targets of known plant miRNAs [43] to further determine whether *BnaAGO*s are regulated by *B. napus* miRNAs. Only *B. napus* miR168a and miR403 were found to have near-perfect complementarity to *BnaAGO1*s and *BnaAGO2*s, respectively (Appendix A, Table 3). Through a phylogenetic analysis, we further identified seven *miR168* orthologs (*ath-miR168a*, *bra-miR168a*, *bna-miR168a*, *ath-miR168b*, *bra-miR168b*, *bna-miR168b*, and *bra-miR168c*) and three *miR403* orthologs (*ath-miR403*, *bra-miR403*, and *bna-miR403*) among the four Brassicaceae species (Appendix A). As far more AGOs than corresponding miRNAs were identified in the four species, we can, therefore, conclude that the miRNAs are more evolutionarily conserved [48,49] than genes such as the *AGO*s.

### 3.2. Diversity and Conservation of the 24 BnaAGOs

The functions of genes are largely determined by their structures [50]. Our comprehensive analyses of the gene structures and physicochemical profiles of the 24 *BnaAGO*s also suggest that *AGO* genes within a subfamily are conserved and are much more similar to each other than to members of other *AGO* subfamilies. The ZIPPY subgroups comprised *AGO* genes with a remarkably smaller gDNA length than those of the other subgroups (Figure 3B, Table 2). Consistently, the ZIPPY subgroup contains only one to five introns, whereas members of the *AGO1*, *AGO4*, and *MEL1* subgroups possess similar gene structure and intron numbers, ranging from 16 to 23 (Figure 3B, Table 2). Furthermore, the gDNA length and intron numbers of the ZIPPY subgroup were also lower than those of homologous genes in *A. thaliana* and *Salvia miltiorrhiza* [28]. These findings suggest that gene structures are conserved within the *AGO* subfamilies, even among different species. On the other hand, some diversity was observed even within subgroups; for example, the structure of *BnaAGO9-3* differed from that of its two paralogs (Figure 3B). Similarly, *BnaAGO2-2* has lost the untranslated region (UTR) present in its three paralogs, resulting in a non-target relationship between *B. napus* miR403 and *BnaAGO2-2* (Appendix A).

The expression patterns also exhibited a similar phenomenon; for instance, genes from the *AGO6*, *AGO7*, and *AGO9* subfamilies were hardly expressed among the 82 tissues studied in total (ZS11, HI, TSW, SPS, and IEN), whereas the *BnaAGO1*s, *BnaAGO2*s, *BnaAGO4*s, and *BnaAGO5*s exhibited significantly higher expression patterns among all the materials (Figure 4 and Figure 5). Contrasting patterns were also observed: *BnaAGO9-3* displayed spatiotemporal expression specificity in the materials, but its paralogs *BnaAGO9-1* and *BnaAGO9-2* were hardly expressed in any of the tissues (Figure 4 and Figure 5). Similarly, *BnaAGO2-2* was hardly expressed in the 64 ZS11 tissues examined, whereas the other three *BnaAGO2*s showed different degrees of spatiotemporal expression specificity (Figure 4). These expression differences may arise from the different gene structures mentioned above (Figure 3B), as suggested previously [50]. In conclusion, the protein characters, gene structures, and expression patterns were highly conserved within the same subfamily while varying significantly across the members of different subfamilies, suggesting that the *AGO* genes underwent frequent gene duplication and recombination during their evolution, which resulted in their functional diversity.

### 3.3. Posttranscriptional Regulation by the B. napus miRNA–AGOs and the Future Regulation of Yield-Related Traits

MicroRNAs are posttranscriptional regulators of gene expression that bind to their specific target mRNAs and load on the RISC to prevent their translation through target cleavage or translation inhibition. The formation of the RISC requires the miRNA to physically interact with AGOs [3]; however, the AGOs and their miRNA triggers remain elusive in *B. napus*. In our study, we predicted two *B. napus* miRNAs mediated by seven *BnaAGO*s through a psRNATarget prediction [43], including four *BnaAGO1*s and three *BnaAGO2*s targeted by mi168a and miR403, respectively (Appendix A, Table 3). A further sequence analysis showed that miR403 perfectly complements the three *BnaAGO2* target sites in their UTRs (Appendix A). In the miR168a–AGO1s, a three-base difference was present in the four target sets (Appendix A). Additionally, the results of the qRT-PCR analysis in eight ZS11 tissues further confirmed the negative expression patterns between miR168a–AGO1s and miR403–AGO2s (Figure 6). Consistently, *AGO1* and *AGO2* were found to be regulated by miR168 and miR403, respectively, in various plants, including *Arabidopsis* [26], rice [27], tomato [51], and *Salvia miltiorrhiza* [28], indicating the conserved regulatory mechanism of the miR168–AGO1 and miR403–AGO2 modules in plants.

In rice, miR168–AGO1 module was not only proven to be helpful in regulating grain yield and flowering time, but also in changing plant immunity to *Magnaporthe oryzae* [27]. In addition, miR403 was shown to play an essential function in regulating tomato development by affecting the expression of *SlAGO2* [51]. To determine whether miR168a–AGO1 and miR403–AGO2 were associated with complex yield-related traits in *B. napus*, we analyzed miRNA-Seq and mRNA-Seq data in multiple genotypes with extremely high and low TSW, SPS, and IEN values. We found that *miR168a* was more highly expressed in low-TSW plants (Figure 7A), whereas *miR403* showed higher expression levels in the high-TSW materials (Figure 7B). Correspondingly, the *BnaAGO1*s (Figure 7A) and *BnaAGO2*s (Figure 7B) displayed the opposite expression patterns to their respective miRNA triggers, indicating that the miR168a–AGO1 module may plays essential negative roles in regulating the TSW trait in *B. napus*, while miR403–AGO2 module have the contrasting function. No significant patterns were detected in the expression patterns of *miR168a*, *miR403*, *BnaAGO1*s, and *BnaAGO2*s between the cultivars with high and low SPS or IEN values, however (Figure 7C–F). Despite this, we cannot conclude that the *B. napus* miR168a–AGO1s and miR403–AGO2 modules were irrelevant to the SPS and IEN character, because the expression of *miR168* and *miR403* have also been proved to reduce the AGO1 and AGO2 protein levels [52]. Furthermore, the AGO2–miR168–AGO1–miR403 loop is known to be vulnerable and easily unbalanced, so even a slight change in this loop would be amplified continuously [51].

## 4. Materials and Methods

### 4.1. Genome-Wide Prediction of AGOs and Their miRNA Triggers

The 10 AtAGOs protein sequences were downloaded from The Arabidopsis Information Resource database (https://www.arabidopsis.org/index.jsp, accessed on 20 June 2020) and used to perform a BLAST analysis to identify the BnaAGOs, BolAGOs, and BraAGOs [53], using an E-value of 0 to identify exact homologs. All the AGO protein sequences identified in the four species were downloaded from the Brassicaceae database (http://brassicadb.cn, accessed on 20 June 2020). MEGA 7.0 was used to perform a multiple sequence alignment using default parameters, and the neighbor-joining method was used to construct the phylogenetic tree with a bootstrap analysis of 1000 replicates [54]. The syntenic analysis was performed using TBtools (https://github.com/CJ-Chen/TBtools, accessed on 16 September 2020) to determine the syntenic relationships of the *AGO* genes from *B. napus* and the three other species.

### 4.2. Chromosomal Location, Gene Structure and Protein Properties

The chromosomal distribution of the *AGO*s was visualized with Map-Chart2.2 [55]. The gene structures and conserved motifs of the *BnaAGO*s were analyzed using Gene Structure Display Server (GSDS) [56] and MEME online (http://meme-suite.org/tools/meme, accessed on 20 June 2020) [57,58], respectively, with the maximum number of predicted motifs set to 15 and default settings used for the other parameters. To further clarify the protein properties of the BnaAGOs, the isoelectric points (PI) and molecular weights of the BnaAGOs were also predicted based on the ExPASy proteomics server database (https://www.expasy.org/tools/, accessed on 20 June 2020) [59].

### 4.3. Identification of B. napus miRNAs with Perfect Complementarity to the BnaAGOs

To select the key *B. napus* miRNAs that target the BnaAGOs, a psRNATarget search (https://www.zhaolab.org/psRNATarget/, accessed on 20 October 2020) was performed using the 24 full-length *B. napus* sequences [43], with the maximum expectation set to 3.5, as described previously [28]. The miRBase database (http://www.mirbase.org/, accessed on 20 October 2020) [60] was searched to identify *miR168* and *miR403* homologs in other species. All miRNA sequences, including the precursor and mature sequences, as well as the regulatory network, were also downloaded from miRBase (http://www.mirbase.org/, accessed on 20 October 2020).

### 4.4. Plant Materials

All seeds of the *B. napus* genotypes with extreme yield-related phenotypes (HI, TSW, SPS, and IEN) were obtained from the Chongqing Rapeseed Technology Research Center, China. All plants were planted in two experimental plots in Chongqing field conditions for at least two years, and ten plants were collected for the character investigation at the harvest stage. Throughout the plant growth, different tissues were obtained and frozen at –80°C until required. The phenotypes of CQ46 (Ningyou 12, low HI) and CQ24 (SWU47, high HI) were published in our previous report [42]. The phenotypes and statistical data for LTSW (P147, low TSW), HTSW (P478, high TSW), LSPS (P026 and P578, low SPS), HSPS (P196 and P446, SPS), LIEN (P355 and P536, low IEN) and HIEN (P300 and P456, high IEN) are presented in Table 4 and Figure 8. The data analysis were performed by a one-way ANOVA method with significant differences (*p* < 0.01).

### 4.5. miRNA-Seq and mRNA-Seq Analysis among Multiple Yield-Related Materials

To examine the expression profiles among the 24 *BnaAGO*s, an analysis was performed on RNA-Seq data from a typical *B. napus* cultivar (ZS11), which can be downloaded from https://brassica.biodb.org/, accessed on 1 October 2022 [41]. Their expression levels in 64 tissues throughout plant growth were used to construct a heatmap. To explore whether the *BnaAGO*s were associated with the complex yield-related traits in *B. napus*, mRNA-Seq data were downloaded for a pair of genotypes with extremely high and low HI, which had been published in our previous study [42]. The expression patterns of the *BnaAGO*s were also explored in another three pairs of yield-related genotypes, with extreme phenotypes in the TSW, SPS, and IEN traits (Appendix A, unpublished data). All heatmaps were constructed using TBtools (https://github.com/CJ-Chen/TBtools, accessed on 16 September 2020). To further clarify the relationships of the *B. napus* miR168a–AGO1s and miR403–AGO2s, and to explore whether the miRNA–AGO modules were associated with the complex yield-related traits in *B. napus*, a further joint analysis of the miRNA-Seq (Appendix A, unpublished data) and mRNA-Seq data (Appendix A, unpublished data) from the TSW, SPS, and IEN materials was performed.

### 4.6. RNA Extraction and qRT-PCR Verification

An RNA extraction kit (Thermo Fisher Scientific, Waltham, MA, USA) was used to extract the total RNA from the specific tissues. A reverse transcription was performed to produce cDNA using a Reverse Transcription Kit (Takara Bio, Kusatsu, Japan). A Bio-Rad CFX system (Bio-Rad Laboratories, Hercules, CA, USA) was used to perform the qRT-PCR analysis in triplicate, and the gene *BnaACTIN* was selected as the endogenous reference for the expression analysis. The 2^−ΔΔCt^ method was selected for the calculation of the relative expression level of the *BnaAGO*s [61]. Finally, the qRT-PCR results were visualized using Graphpad Prism 5.0 [62]. To determine the expression levels of *B. napus miR168a* and *miR403*, the gene *BnaU6* was selected as the endogenous reference. All the primers used in our study were designed with Primer 5.0 [63] and are listed in Appendix A.

## 5. Conclusions

In this work, a comprehensive approach includes the phylogenetic analysis of the AGOs in *A. thaliana*, *B. napus*, *B. rapa*, and *B. oleracea*, the gene structures, conserved motifs, and expression patterns of the 24 *BnaAGO*s in many differently yielding genotypes. Our results reveal the diversity and conservation of the sequences and functions among these genes, as well as indicating the loss of some orthologs following the cross that resulted in *B. napus*. In addition, a posttranscriptional regulation analysis indicated that the miR168a–AGO1 and miR403–AGO2 regulatory modules previously reported in other species were conserved in *B. napus* and may play roles in regulating yield-related traits in this oilseed crop, especially in TSW. Our findings, therefore, provide a new understanding of the miR168–AGO1s and miR403–AGO2s modules of *B. napus*, facilitating future research into the complex yield traits.

## Figures and Tables

**Figure 1 ijms-24-02543-f001:**
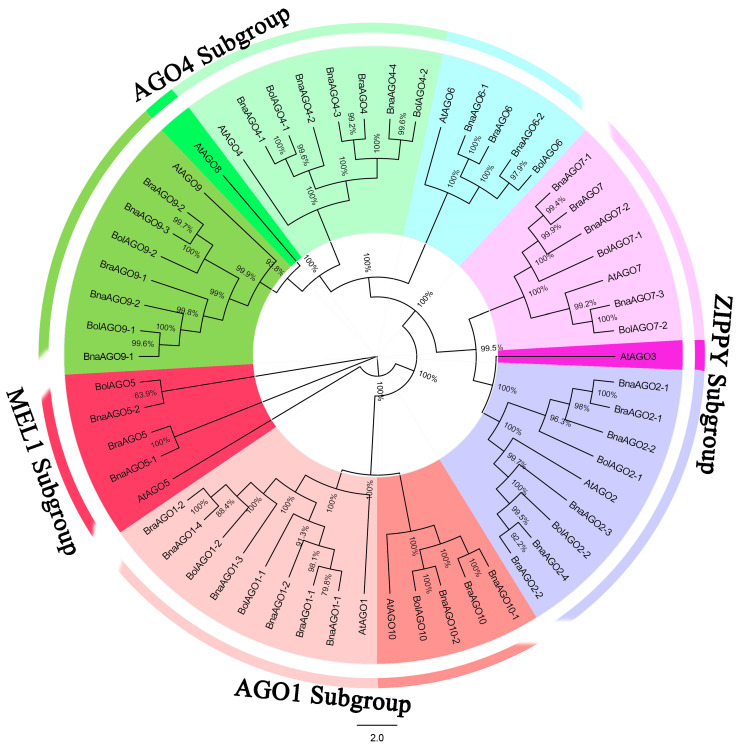
Neighbor-joining (NJ) tree of the Argonaute (AGO) protein sequences from *B. napus*, *B. oleracea*, *B. rapa* and *A. thaliana*. All the AGO genes were renamed from AGO1 to AGO10.

**Figure 2 ijms-24-02543-f002:**
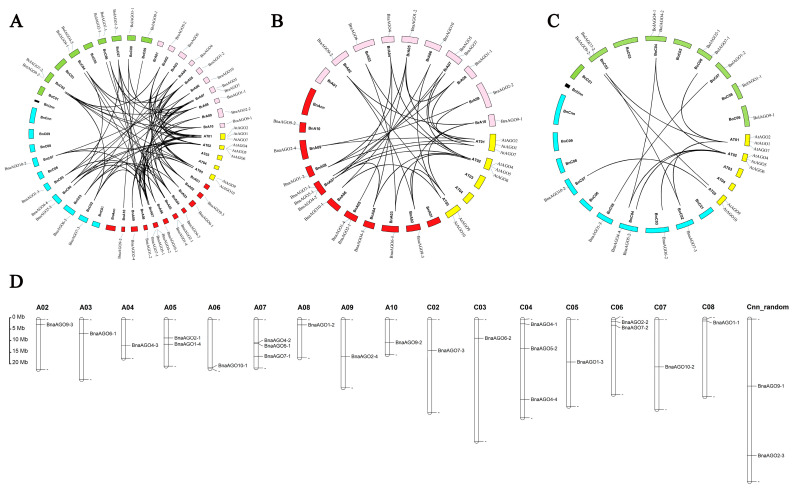
Syntenic relationships of AGO genes (**A**–**C**) and the chromosome location of the 24 *BnaAGOs* (**D**). (**A**) Syntenic relationships of the total 58 AGOs among the 4 species. (**B**) Syntenic relationships of AGOs among the *A. thaliana*, *B. rapa* and A-subgenomic from *B. napus*. (**C**) Syntenic relationships of AGOs among the *A. thaliana*, *B. oleracea* and C-subgenomic from *B. napus*.

**Figure 3 ijms-24-02543-f003:**
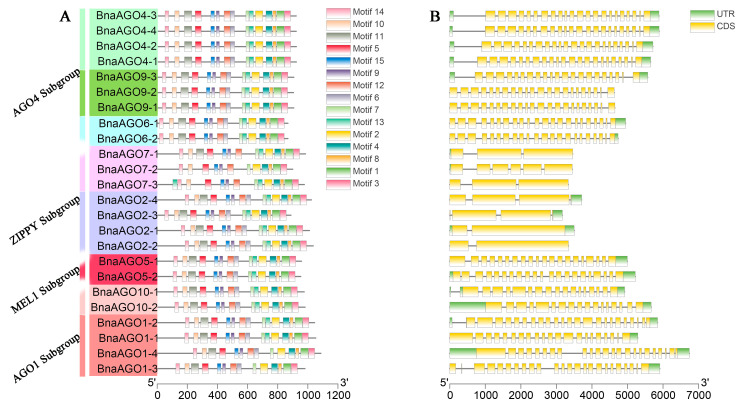
The conserve motif (**A**) and gene structure (**B**) analysis among the 24 *BnAGOs*.

**Figure 4 ijms-24-02543-f004:**
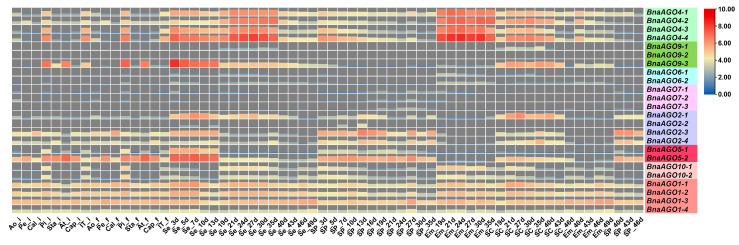
The expression level of the 24 identified BnaAGOs in ZS11 with 64 different tissues. The figure of 3d, 5d, 7d, 10d, 13d, 19d, 21d, 24d, 27d, 30d, 35d, 40d, 43d, 46d and 49d on the X-abscissa means the days after flowering; “i” indicate the initial flowering stage; “f” represent the flowering stage with full-bloom; Ao: anthocaulus; Pe: petal; Cal: calyx; Pi: pistil; Sta: stamen; At: anther; Cap: capillament; IT: inflorescence top; Se: seed; SP: silique pericarp; Em: embryo; SC: seed coat.

**Figure 5 ijms-24-02543-f005:**
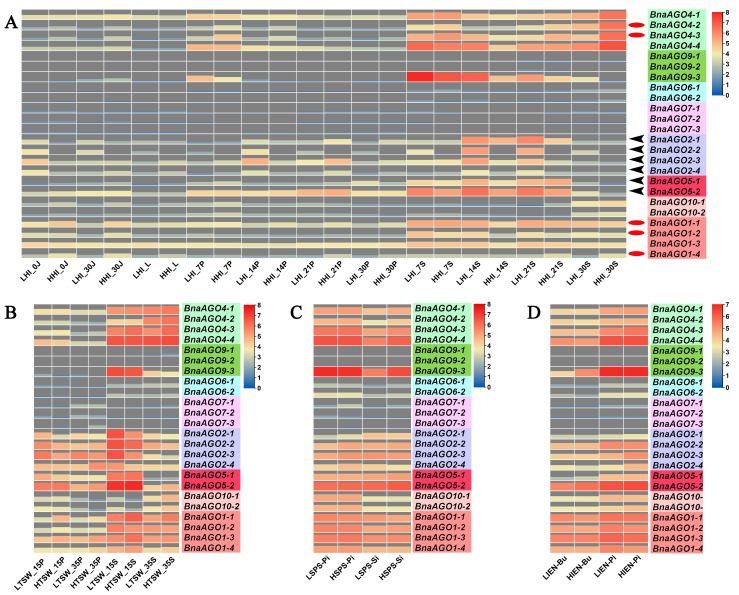
The expression level of the 24 identified *BnaAGO*s in yield-related materials with differences in harvest index (HI); (**A**), thousand-seed weight (TSW;) (**B**), seed number per silique (SPS); (**C**), and initial embryonic number (IEN;) (**D**). The numbers used in the labels on the *x*-axis (0, 7, 10, 14, 15, 20, 21, 30, and 35) indicate the day after flowering; LHI and HHI indicate high- and low-HI materials; LTSW and HTSW indicate high- and low-TSW materials; LSPS and HSPS indicate high- and low-SPS materials; LIEN and HIEN indicate high- and low-IEN materials; J: stem; L: leaf; Bu: bud; P: pericarp; S: seed; Pi: pistil; Si: silique.

**Figure 6 ijms-24-02543-f006:**
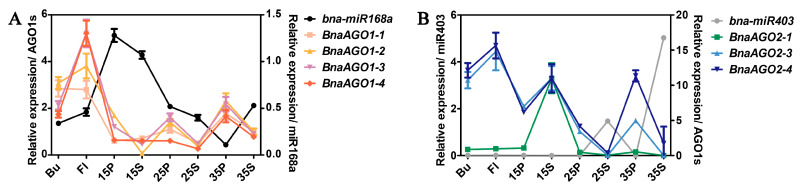
The qRT-PCR verification of the expression pattern between *B. napus* miR168a-AGO1s (**A**) and miR403-AGO2s (**B**) in materials ZS11. The numbers used in the labels on the *x*-axis (15, 25, and 35) means the days after flowering; Bu: bud; Fl: flower; P: pericarp; S: seed.

**Figure 7 ijms-24-02543-f007:**
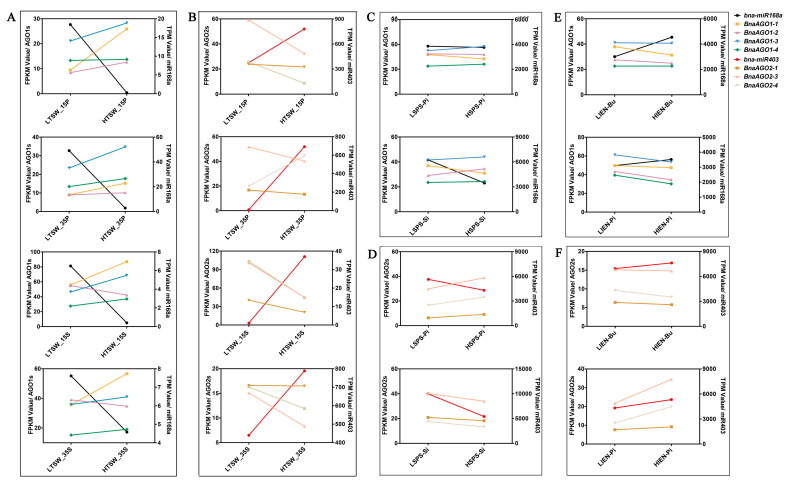
The joint analysis between miRNA-Seq and mRNA-Seq of *B. napus* miR168-AGO1s and miR403-AGO2s in yield-related materials with differences in thousand-seed weight (TSW); (**A**,**B**), seed number per silique (SPS); (**C**,**D**), and initial embryonic number (IEN); (**E**,**F**). LTSW and HTSW indicate high- and low-TSW materials; LSPS and HSPS indicate high- and low-SPS materials; LIEN and HIEN indicate high- and low-IEN materials; FPKM: Fragments Per Kilobase of exon model per Million mapped fragments; TPM: Transcripts Per Kilobase of exon model per Million mapped reads; The numbers used in the labels on the *x*-axis (15 and 35) indicate the day after flowering; P: pericarp; S: seed; Pi: pistil; Si: silique; Bu: bud. All FPKM and TPM data were shown in Appendix A, respectively.

**Figure 8 ijms-24-02543-f008:**
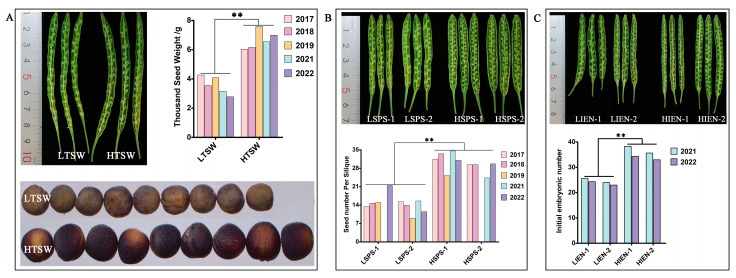
Phenotype and statistical analysis of the materials with differences in thousand-seed weight (TSW; **A**), seed number per silique (SPS; **B**), and initial embryonic number (IEN; **C**) in *B. napus*. LHI and LTSW and HTSW indicate high- and low-TSW materials; LSPS and HSPS indicate high- and low-SPS materials; LIEN and HIEN indicate high- and low-IEN materials; The *p* value were performed with one-way ANOVA and ** represent significant difference at *p*_Value < 0.01.

**Table 1 ijms-24-02543-t001:** The copy number of AGOs among *A. thaliana*, *B. rapa*, *B. oleracea*, *B. napus* with A and C sub-genomes.

Gene Name	*A. thaliana*	*B. rapa*	*B. naus* (A)	*B. oleracea*	*B. naus* (C)
Zero-copy in *B. napus*					
AGO3	1	0	0	0	0
AGO8	1	0	0	0	0
Two-copy in *B. napus*					
AGO5	1	1	1	1	1
AGO10	1	1	1	1	1
AGO6	1	1	1	1	1
Three-copy in *B. napus*					
AGO7	1	1	1	2	2
AGO9	1	2	2	2	1
Four-copy in *B. napus*					
AGO1	1	2	2	2	2
AGO2	1	2	2	2	2
AGO4	1	1	2	2	2

**Table 2 ijms-24-02543-t002:** Complete list of the 24 *BnaAGOs* identified in our study.

Gene Name	Transcript Name	gDNA Size (bp)	CDS Size (nts)	Protein			No. of Introns	Genomic Location
				Length (aa)	Mw (kDa)	pI		
*BnaAGO1-1*	*BnaC08g46720D*	5291	3159	1052	116.78	9.45	20	chrC08_random:953598-958888
*BnaAGO1-2*	*BnaA08g03260D*	5847	3135	1044	115.85	9.4	22	chrA08:2681072-2686918
*BnaAGO1-3*	*BnaC05g25730D*	5910	2943	980	100.15	9.17	22	chrC05:21116236-21122145
*BnaAGO1-4*	*BnaA05g17460D*	6740	3261	1086	120.33	9.38	22	chrA05:12290150-12296889
*BnaAGO2-1*	*BnaA05g14760D*	3501	3033	1010	112.74	9.51	1	chrA05:9228612-9232112
*BnaAGO2-2*	*BnaC06g41790D*	3340	3108	1035	114.35	9.54	1	chrC06_random:1066401-1069740
*BnaAGO2-3*	*BnaCnng68320D*	3169	2664	887	100.7	9.44	2	chrCnn_random:67905947-67909115
*BnaAGO2-4*	*BnaA09g25290D*	3712	3072	1023	113.91	9.66	3	chrA09:18324937-18328648
*BnaAGO4-1*	*BnaC04g54830D*	5652	2772	923	103.36	8.87	21	chrC04_random:2230442-2236093
*BnaAGO4-2*	*BnaA07g13010D*	5716	2772	923	103.36	8.82	21	chrA07:11653377-11659092
*BnaAGO4-3*	*BnaA04g15560D*	5887	2769	922	103.06	8.91	21	chrA04:12884285-12890171
*BnaAGO4-4*	*BnaC04g38560D*	5890	2769	922	103.11	8.87	21	chrC04:39739228-39745117
*BnaAGO5-1*	*BnaA07g13430D*	4997	2874	957	106.71	9.52	19	chrA07:11907998-11912994
*BnaAGO5-2*	*BnaC04g16450D*	5220	2859	952	106.03	9.62	20	chrC04:14487678-14492897
*BnaAGO6-1*	*BnaA03g15180D*	4948	2604	867	97.2	9.03	21	chrA03:7005357-7010304
*BnaAGO6-2*	*BnaC03g18310D*	4736	2604	867	97.37	8.99	21	chrC03:9391663-9396398
*BnaAGO7-1*	*BnaA07g24280D*	3461	2955	984	112.47	9.37	2	chrA07:18160385-18163845
*BnaAGO7-2*	*BnaC06g43420D*	3447	2700	899	102.67	9.38	5	chrC06_random:2865213-2868659
*BnaAGO7-3*	*BnaC02g19190D*	3334	2931	976	111.58	9.32	2	chrC02:15451981-15455314
*BnaAGO9-1*	*BnaCnng35060D*	4647	2721	906	102.57	9.42	21	chrCnn_random:33265084-33269730
*BnaAGO9-2*	*BnaA10g14450D*	4627	2715	904	102.04	9.31	21	chrA10:11492941-11497567
*BnaAGO9-3*	*BnaA02g05290D*	5571	2721	906	101.34	9.31	20	chrA02:2403187-2408757
*BnaAGO10-1*	*BnaA06g36540D*	4921	2928	975	109.27	9.38	16	chrA06:23915363-23920283
*BnaAGO10-2*	*BnaC07g17330D*	5667	2946	981	109.75	9.38	16	chrC07:23533982-23539648

**Table 3 ijms-24-02543-t003:** The complementarities between *B. napus* miRNAs and the 7 *BnaAGOs*.

miRNA_ID	Target_Name	Expection	miRNA_aligned_fragment	Target_aligned_fragment	Alignment	Inhibition
*bna-miR403*	*BnaAGO2-4*	0	UUAGAUUCACGCACAAACUCG	GGAGUUUGUGCGUGAAUCUAA	::::::::::::::::::::	Cleavage
*bna-miR403*	*BnaAGO2-1*	0	UUAGAUUCACGCACAAACUCG	GGAGUUUGUGCGUGAAUCUAA	::::::::::::::::::::	Cleavage
*bna-miR403*	*BnaAGO2-3*	0	UUAGAUUCACGCACAAACUCG	GGAGUUUGUGCGUGAAUCUAA	::::::::::::::::::::	Cleavage
*bna-miR168a*	*BnaAGO1-2*	3	UCGCUUGGUGCAGGUCGGGAA	UUCCCGAGCUGCAUCAAGCUA	::::::: :::::.::::: :	Cleavage
*bna-miR168a*	*BnaAGO1-1*	3	UCGCUUGGUGCAGGUCGGGAA	UUCCCGAGCUGCAUCAAGCUA	::::::: :::::.::::: :	Cleavage
*bna-miR168a*	*BnaAGO1-4*	3	UCGCUUGGUGCAGGUCGGGAA	UUCCCGAGCUGCAUCAAGCUA	::::::: :::::.::::: :	Cleavage
*bna-miR168a*	*BnaAGO1-3*	3	UCGCUUGGUGCAGGUCGGGAA	UUCCCGAGCUGCAUCAAGCUA	::::::: :::::.::::: :	Cleavage

**Table 4 ijms-24-02543-t004:** Phenotypic analysis of yield-related extremely materials during multiple years of field tests.

Trait	Material	2016	2017	2018	2019	2021	2022	Mean Value	SEM	*p*_Value
Thousand Seed Weight/g (TSW)	LTSW	-	4.27	3.54	4.08	3.15	2.78	3.56	0.56	
	HTSW	-	6.03	6.15	7.58	6.55	6.99	6.66	0.57	0.000
Seed number Per Silique (SPS)	LSPS-1	-	13.47	14.74	14.99	-	21.50	16.17	3.13	
	LSPS-2	-	15.30	13.97	8.98	15.59	11.50	13.07	2.51	
	HSPS-1	-	31.43	33.53	25.23	34.60	31.00	31.16	3.25	0.00
	HSPS-2	-	29.40	29.40	-	24.36	29.70	28.22	2.23	
Initial embryonic number (IEN)	LIEN-1	-	-	-	-	25.67	24.33	25.00	0.67	
	LIEN-2	-	-	-	-	24.00	23.00	23.50	0.50	
	HIEN-1	-	-	-	-	38.33	34.33	36.33	2.00	0.00
	HIEN-2	-	-	-	-	35.67	33.00	34.34	1.34	

## Data Availability

Not applicable.

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
