# Peer review of "Genome-Wide Identification and Posttranscriptional Regulation Analyses Elucidate Roles of Key Argonautes and Their miRNA Triggers in Regulating Complex Yield Traits in Rapeseed"

_ijms, 2023, doi:10.3390/ijms24032543_

Round 1

Reviewer 1 Report

Presented study concerns an economically important species, Brassica napus. The aim of the study is to identify rapeseed proteins that belong to Argonautes family and their relationship with miRNAs.

Manuscript construction is clear. The introduction provides good background and justification of the presented study. The presentation of the results does not raise any objections as well. The conclusions are adequate to the presented results. Most of the methods are indisputable. However, I have some concerns I would like the Authors to explain.

Genome-Wide identification of AGOs in B. napus has been already done by Cao et al. 2016. However, in the presented research I don't see any reference to that study and it's not clear to me what is the novelty of the results, regarding the identification of AGOs. Moreover, why Neighbor-Joining method was chosen rather than Maximum Likelihood, which is widely accepted for the phylogenetic analysis? Also, in my opinion model for testing phylogenetic relationship should be chosen based on Bayesian information criterion. 

Reviewer 2 Report

I think that manuscript titled: Genome-wide identification and posttranscriptional regulation analyses elucidate roles of key Argonautes and their miRNA triggers in regulating complex yield traits in rapeseed addresses important issues such as a better understanding of the factors regulating the yield of plants is crucial for the development of agriculture. The manuscript is well-planned and prepared. Tet is consistent and easy to follow. However, in the chapter material and methods  there is lacking information about the method of statistical analysis. This information is crucial. without this information, it is difficult to assess whether appropriate statistical methods have been used. Also, the captions under the figures lack such information because there is only this information (Line 427):**: significant difference at P Value < 0.01. There is no information on what kind of test was applied.

Therefore presented manuscript could be accepted for publication in International Journal of Molecular Science after minor revision and updating information about statistical analysis.
